# Imbalance of Drug Transporter-CYP450s Interplay by Diabetes and Its Clinical Significance

**DOI:** 10.3390/pharmaceutics12040348

**Published:** 2020-04-11

**Authors:** Yiting Yang, Xiaodong Liu

**Affiliations:** Center of Drug Metabolism and Pharmacokinetics, China Pharmaceutical University, Nanjing 210009, China; 1821010209@stu.cpu.edu.cn

**Keywords:** diabetes, transporter-enzyme interplay, influx transporter, efflux transporter, physiologically based pharmacokinetic model, pharmacokinetics, cytochrome P450 enzymes

## Abstract

The pharmacokinetics of a drug is dependent upon the coordinate work of influx transporters, enzymes and efflux transporters (i.e., transporter-enzyme interplay). The transporter–enzyme interplay may occur in liver, kidney and intestine. The influx transporters involving drug transport are organic anion transporting polypeptides (OATPs), peptide transporters (PepTs), organic anion transporters (OATs), monocarboxylate transporters (MCTs) and organic cation transporters (OCTs). The efflux transporters are P-glycoprotein (P-gp), multidrug/toxin extrusions (MATEs), multidrug resistance-associated proteins (MRPs) and breast cancer resistance protein (BCRP). The enzymes related to drug metabolism are mainly cytochrome P450 enzymes (CYP450s) and UDP-glucuronosyltransferases (UGTs). Accumulating evidence has demonstrated that diabetes alters the expression and functions of CYP450s and transporters in a different manner, disordering the transporter–enzyme interplay, in turn affecting the pharmacokinetics of some drugs. We aimed to focus on (1) the imbalance of transporter-CYP450 interplay in the liver, intestine and kidney due to altered expressions of influx transporters (OATPs, OCTs, OATs, PepTs and MCT6), efflux transporters (P-gp, BCRP and MRP2) and CYP450s (CYP3As, CYP1A2, CYP2E1 and CYP2Cs) under diabetic status; (2) the net contributions of these alterations in the expression and functions of transporters and CYP450s to drug disposition, therapeutic efficacy and drug toxicity; (3) application of a physiologically-based pharmacokinetic model in transporter–enzyme interplay.

## 1. Introduction

The pharmacokinetics of a drug is determined by absorption, distribution, excretion and metabolism. Drug absorption, distribution and excretion are mainly controlled by drug transporters, while drug metabolism is mediated by metabolic enzymes. These drug transporters are classified into the ATP binding cassette (ABC) family and the solute carrier (SLC) family. The main SLC transporters involved in drug transport are multidrug/toxin extrusions (MATEs), organic anion transporting polypeptides (OATPs), monocarboxylate transporters (MCTs), organic anion transporters (OATs), peptide transporters (PepTs), and organic cation transporters (OCTs). Most SLC transporters belong to influx transporters, except for MATEs. The identified ABC transporters related to drug efflux include P-glycoprotein (P-gp), multidrug resistance-associated proteins (MRPs) and breast cancer resistance protein (BCRP). These transporters are widely expressed in the intestine, liver and kidney. They affect drug therapeutic effects/toxicity via regulating drug uptake or secretion. Enzymes involved in drug metabolism mainly include cytochrome P450 enzymes (CYP450s) and UDP-glucuronosyltransferases (UGTs). These enzymes are also widely distributed in the liver, intestine and kidney. Drug disposition in tissues is highly dependent on the coordinate work of these influx transporters, enzymes and efflux transporters, termed as the “interplay of transporters and enzymes” [1,2,3]. Moreover, multiple SLC transporters, ABC transporters and enzymes often participate in the disposition of drugs. A typical example is atorvastatin. Atorvastatin is a substrate of P-gp, MRP2, BCRP, OATP1B1, OATP1B2, OATP2A1, CYP3A4/5 [4] and UGTs [5,6]. Sodium taurocholate co-transporting polypeptide (NTCP) also mediates hepatic uptake of atorvastatin [7]. In the liver, atorvastatin enters hepatocytes from portal blood mainly via influx transporters at the basolateral surface of hepatocytes. In hepatocytes, atorvastatin is metabolized via CYP3As or UGTs. Atorvastatin and its metabolites are excreted into bile via efflux transporters (MRP2, BCRP and P-gp) or returned to the blood via MRP3 [8]. The transporter–enzyme interplay also occurs in the intestine and kidney. Accumulating studies [9,10,11,12] have demonstrated that diabetes remarkably alters the expression and functions of drug transporters and CYP450s, disordering transporter-CYP450 interplay and in turning affecting the disposition of corresponding drugs, their therapeutic efficacy or drug toxicity. Here, we aimed to focus on (1) the imbalance of transporter-CYP450 interplay in the liver, intestine and kidney due to alterations in the expression of influx transporters, efflux transporters and CYP450s under diabetic status; (2) the net contributions of the altered expressions and functions of transporters and CYP450s to drug disposition, therapeutic efficacy and drug toxicity; (3) application of a physiologically based pharmacokinetic model (PBPK) in the imbalance of transporter–enzyme interplay under diabetic conditions.

## 2. Liver

Drugs are eliminated in the liver mainly via metabolism and biliary excretion. The liver highly expresses various drug transporters (such as P-gp, BCRP, MRPs, OATPs, OCTs, OATs and MATEs) (Figure 1A) and drug metabolic enzymes (such as CYP450s and UGTs). They work in series to control the disposition of drugs in the liver (Figure 1B).

### 2.1. OATPs

OATPs, expressed at the basolateral membrane of hepatocytes, mediates hepatocyte uptake of many anions from the blood, which becomes a rate-limited process of hepatic clearances for some drugs [3]. It was found that in diabetic rats (type 2 diabetes) induced by a streptozocin (STZ) and high-fat diet (HFD) combination (termed as STZ/HFD), hepatic OATP1B2 (rodent orthologue of human OATP1B1 and OATP1B3) is remarkably induced, leading to increased hepatic uptake of repaglinide [9,10], atorvastatin [9,11] and simvastatin [12] as well as hepatic clearances [9,11]. In line with this, the diabetic rats showed lower plasma concentrations of atorvastatin [9,11], simvastatin [12] and pravastatin [13] compared with control rats. Clinical trials also showed that tuberculosis patients with diabetes showed significantly lower concentrations of rifampicin than non-diabetic patients [14,15], which may be attributed to the induction of hepatic OATPs by diabetes and rifampicin.

### 2.2. P-gp

P-gp, expressed at the canalicular membrane of hepatocytes, mediates biliary excretion of its substrates. Diabetes was reported to upregulate the expression of hepatic Mdr1b (ABCB1b) mRNA and P-gp protein in 8-day diabetic Wistar rats (type 1 diabetes) induced by STZ, which was associated with the activation of protein kinase C alpha (PKCα) and nuclear factor kappa-B (NF-κB) [16]. A similar increase in the expression of hepatic Mdr1b mRNA was found in STZ-induced diabetic rats [17]. The induction of hepatic P-gp protein expression was also observed in STZ/HFD-induced diabetic rats [11]. However, a report showed that in STZ-induced diabetic Sprague-Dawley rats, 5-week diabetes significantly increased Mdr1a (ABCB1a) and decreased Mdr1b mRNA expression, but significantly decreased expressions of both Mdr1a and Mdr1b mRNA were found in 8-week diabetic rats. Immunoblotting demonstrated that 5-week diabetes significantly reduced the expression of P-gp protein in rats, but the decreases were not found in 8-week diabetic rats, indicating that the regulation of P-gp protein occurred at post-transcriptional level under diabetic status [18].

### 2.3. BCRP

BCRP, expressed at the canalicular membrane of hepatocytes, is also involved in the biliary excretion of its substrates. Altered expressions of hepatic BCRP under diabetic status are dependent on the type of diabetes, and diabetic progression. For example, expressions of hepatic BCRP mRNA in 5-week and 8-week diabetic rats induced by STZ were significantly downregulated, but expression of BCRP protein only showed a trend to decrease [19]. Similarly, He et al. reported that hepatic BCRP mRNA level was decreased in STZ-induced diabetic rats, whereas in type 2 diabetic patients and Goto-Kakizaki (GK) rats BCRP mRNA expression was significantly increased, although BCRP protein was unaltered [20]. Decreased expression of hepatic BCRP protein was observed in 10-day (but not 22-day) diabetic rats induced by STZ/HFD [11]. Interestingly, pregnancy also affected the expression of hepatic BCRP under diabetic status. For example, STZ treatment upregulated the expression of BCRP mRNA and protein in non-pregnant mice, but not in pregnant mice [21].

### 2.4. MRP2

MRP2 is co-expressed with P-gp and BCRP at the canalicular membrane of hepatocytes. Alterations in the expression of MRP2 by diabetes are also dependent on species. STZ-induced diabetic Donryu rats showed significantly lower expression of hepatic MRP2 mRNA, which was in line with lower biliary excretion of pravastatin [13]. A similar decrease in the protein (not mRNA) of hepatic MRP2 was found in STZ-induced diabetic male Wistar rats [22]. However, STZ-induced diabetic Sprague-Dawley rats showed higher expression of hepatic MRP2 protein, which was in line with higher biliary excretion of sulfobromophthalein compared with control rats [23]. Moreover, STZ treatment upregulated the expression of MRP2 in non-pregnant mice but not in pregnant mice [21]. The induction of hepatic MRP2 may partly explain the increased biliary excretions of cefoperazone and some anions (such as bengal, indocyanine green and bromcresol green) under diabetic status [24,25].

### 2.5. CYP450s

Growing evidence has demonstrated that diabetes increases the expression of some CYP450s (such as CYP2A1, CYP1A2, CYP2B1/2, CYP2C6, CYP2C7, CYP3A1/2 and CYP2E1) in rats [9,11,12,26,27,28,29,30], although opposite reports have also been shown [31,32]. The induction of CYP3A1/2 led to lower plasma exposure of simvastatin [12], atorvastatin [9,11], verapamil [27,29] and lidocaine [30] following intravenous dose to diabetic rats. The formation of 1,3-dimethyluric acid from theophylline is mainly mediated via CYP1A2 and CYP2E1. Both alloxan-induced diabetes and STZ-induced diabetes were reported to increase expressions of hepatic CYP2E1 and CYP3A2 by three-fold, leading to a lower area under the curve (AUC) of theophylline and a higher AUC of 1,3-dimethyluric acid following oral and intravenous administration to rats [28]. The induction of CYP1A2 (and CYP3A1/2) may contribute to a lower AUC of oltipraz [33], higher non-renal clearances of omeprazole [34] and systemic clearance of antipyrine [35] in diabetic rats. Diazinon is metabolized to toxic metabolite diethylphosphate via CYP3A2 and CYP1A2. The increases in diazinon toxicity and urinary recovery of its metabolite diethylphosphate in diabetic rats may be attributed to the enhancement of hepatic CYP1A2 (and CYP3A2)-mediated metabolism of diazinon [35]. A report showed that protein expression and activity of hepatic CYP2E1 in STZ-induced diabetic rats were increased by three- and 2.5-fold in comparison to control rats, respectively. The increases were in line with the increased oxidative stress, demonstrating crucial roles of CYP2E1 in stress-induced pathological processes under diabetic status [31,32,36].

Hyperketonemia (such as acetone) is considered as a reason for diabetes inducing CYP2E1. Data from primary cultured rat hepatocytes demonstrated that insulin decreased the expression of CYP2E1 and CYP2B protein and mRNA, inferring that a deficiency of insulin is also a reason that diabetes induces the expression of CYP2El and CYP2B [37]. Furthermore, data from both Fa2N-4 cells and HepG2 cells demonstrated that fatty acids (palmitic acid, oleic acid, stearic acid and linoleic acid), but not insulin, upregulated the activity and expression of CYP3A4 mRNA and protein, indicating that increased levels of fatty acids may be one of the reasons that diabetes elevated the function and expression of CYP3A4 [38]. Recent studies showed that diabetes reduced the expression of hepatic peroxisome proliferator activated receptor γ (PPARγ) protein [39,40] and PPARα protein [41]. The decreased expression of hepatic PPARα was attributed to the increased palmitic acid [41]. A report showed that PPARα agonist gemfibrozil in PPARα-null mice showed more induction of the mRNA, protein and activity of CYP3a, CYP2b and CYP2c than in wild-type mice, indicating that the induction of gemfibrozil on CYP450s was suppressed by PPARα activation [42]. The downregulation of PPARα protein expression by diabetes [41] might attenuate the inhibition of PPARα on the expression of CYP450s, leading to the induction of hepatic CYP450s, which needs further investigation. Moreover, diabetes was reported to increase the expression of hepatic glucocorticoid receptor and increase the circulating level of corticosterone [43,44]. The increased level of corticosterone might induce expressions of CYP2B [45] and CYP2C [46] via activating glucocorticoid receptor. All these results demonstrate that alterations in the expressions of hepatic CYP450s were involved in various mechanisms. In contrast to CYP2C6 induction, diabetes significantly lowered the expression of CYP2C11 [26,47,48,49,50], impairing the metabolism of diclofenac [49], glibenclamide [48] and nateglinide [47]. The increase in CYP2C6 expression and decrease in CYP2C11 expression may partly explain why the AUC values of both phenytoin and 4′-hydroxylphenytoin in diabetic rats were comparable to control rats [51].

Some contradictory results have also been reported. In Goto-Kakizaki rats, it was found that expressions of hepatic CYP reductase and CYP3A2 were significantly upregulated, accompanied by increases in the activities of midazolam 4-hydroxylase and CYP reductase. In contrast, hepatic expressions of CYP1A2 and CYP3A1 were downregulated, and the activities of 7-methoxyresorufin-*O*-demethylase and 7-ethoxyresorufin-O-deethylase were decreased. Expressions of other CYP450s, such as CYP1B1, CYP2B1, CYP2C11 and CYP2E1, were unaltered, inferring that diabetes regulates the expression of hepatic CYP450s in an isoform-specific manner [30,52]. Similarly, in 20-week-old male db/db mice, testosterone-6β-hydroxylation but not midazolam-1-hydroxylation was significantly decreased, the intrinsic clearance (CL_int_) value of testosterone-6β-hydroxylation (CYP3A11) was less than 46% of control mice. Other metabolic reactions such as dextromethorphan-*O*-demethylation (CYP2D), phenacetin-*O*-deethylation (CYP1A2), coumarin-7-hydroxylation (CYP2A4/5), tolbutamide-4-hydroxylation (CYP2C), bupropion-hydroxylation (CYP2B10) and omeprazole-5-hydroxylation (CYP2C29), chlorzoxazone-6-hydroxylation (CYP2E1) were unaltered [53]. Expressions and functions of hepatic CYP450s are dependent on diabetic progression. It was reported that 25-week-old db/db mice showed substantial decreases in mRNA expressions and functions of CYP2B10 and CYP2C29 compared with those of the 10-week-old db/db mice. But mRNA expressions of CYP3A11 and CYP2E1 in the 25-week-old db/db mice were comparable to those of the 10-week-old db/db mice [54]. In Zucker diabetic fatty (ZDF) rats, CYP1A1/2, and CYP3A1 levels were similar among the livers of 5 and 11-week-old ZDF rats and control rats, although 11-week-old ZDF rats possessed the higher serum glucose and glycated hemoglobin levels, but not 5-week-old ZDF rats. Moreover, 5-week-old ZDF rats, but not 11-week-old ZDF rats, showed lower levels of cytochrome b5, CYP2B1, CYP2C11, CYP2E1 and CYP3A2. Consistently, pentoxyresorufin O-depentylation, testosterone 2α- and 16α-hydroxylation, chlorzoxazone 6-hydroxylation, and midazolam 1’- and 4-hydroxylation were decreased only in 5-week-old ZDF rats [55]. Expressions (mRNA and proteins) of CYP3A11/13 and their activities were significantly increased in STZ-induced diabetic C57BLKS/J mice and db/db mice [56]. In contrast, Wang et al. reported that the levels of hepatic CYP3A11 mRNA in C57BL/6 mice diabetic mice induced by HFD/STZ were less than 30% of control mice [31]. All these results indicate that alterations in the expressions of hepatic CYP450s are highly dependent on the type of diabetes, diabetic progression and animal species.

Expressions of hepatic CYP450s in diabetes patients have been widely investigated, although some results are often in contrast to findings in diabetic mice. Gravel et al. [57] assessed the activities of seven major CYP450s (CYP1A2, CYP2B6, CYP2C9, CYP2C19, CYP2D6, CYP2E1, and CYP3A4/5) in 38 type 2 diabetic patients and 35 non-diabetic subjects following the oral administration of a cocktail of probes (caffeine, bupropion, tolbutamide, omeprazole, dextromethorphan, chlorzoxazone and midazolam). They found that the activities of CYP2C19, CYP2B6, and CYP3A in diabetic patients were decreased by about 46%, 45%, and 38%, respectively. CYP1A2 and CYP2C9 activities showed a trend to slightly increase in subjects with diabetes, but activities of CYP2D6 and CYP2E1 were unaltered [57]. Meanwhile, obese children showed higher oral clearance of chlorzoxazone and lower AUC of chlorzoxazone. The AUC was only 46% of nonobese peers, inferring the increased CYP2E1 activity [58].

In vitro data demonstrated that protein expressions and activities of CYP3A4 in hepatic microsomes of diabetic livers were significantly lower than those in non-diabetic donors, although a great variability (8.2-fold) in CYP3A4 protein levels was observed. However, the expression and activity of CYP2E1 were significantly increased [59]. The disposition of nisoldipine in humans was investigated. The results show that the disposition of nisoldipine is enantioselective and nisoldipine is preferentially metabolized. Compared with non-diabetes patients, AUC values of (+)-nisoldipine and (−)-nisoldipine in diabetes patients were increased by 94% and 143%, respectively. Similar increases were also found in their peak concentration (C_max_). Lidocaine tests demonstrated a decrease in activity of CYP3A (and CYP1A2) [60]. Similarly, pregnant women with gestational diabetes showed significantly higher C_max_ and AUC of lidocaine following peridural anesthesia administration of lidocaine than non-diabetic patients [61]. However, oral plasma concentrations of nifedipine [62] and metoprolol [63] in pregnant women with gestational diabetes were comparable to those in non-diabetic pregnant women. Adith et al. [64] investigated phenytoin kinetics in 10 male type 1 and 10 type 2 diabetic patients. Age- and sex-matched epileptic patients receiving phenytoin alone served as control groups. They found that steady-state concentrations of phenytoin were significantly lower in both types of diabetic patients compared to the respective controls. In line with this, diabetes patients showed significantly increased V_max_/K_m_ values of phenytoin. Similarly, the AUCs of cyclosporin A in kidney transplant recipients with diabetes mellitus were approximately 50%~60% that of non-diabetic patients [65,66], which was partly attributed to the delayed gastric emptying [66]. Moreover, no difference in cyclosporine A trough levels between renal transplant patients and renal transplant patients with diabetes was found [67]. Diabetes patients were also reported to show a trend to increase in the expression and activity of CYP3A4, although no statistical significance was observed. Importantly, significant decreases were found in diabetic patients with non-alcoholic fatty liver or non-alcoholic steatohepatitis. Furthermore, the decrease in activity and protein level of CYP3A4 continued with the severity of disease as it progressed from non-alcoholic fatty liver to non-alcoholic steatohepatitis [68]. Matzke et al. [69] also investigated the activities of CYP1A2 and CYP2D6 in type 1 diabetes and type 2 diabetes patients using the probes antipyrine, caffeine and dextromethorphan. The results show that compared with controls, oral clearance of antipyrine was increased 72% in type 1 diabetes patients. Formation clearances of 4-hydroxyantipyrine and 3-hydroxymethylantipyrine were increased by 74% and 137%, respectively. The caffeine metabolic index (paraxanthine/caffeine) was increased by 34%, but no statistical significance was obtained. These alterations did not occur in type 2 diabetes patients. Similarly, although plasma clearance of theophylline in diabetic patients was comparable to that in healthy subjects, there was a positive correlation between hemoglobin A1c values and formation clearances of both 1,3-dimethyluric acid and 1-methyluric acid [70], in line with inductions of CYP1A2 and CYP2E1.

Although antipyrine is widely used to assess the activity of CYP450s in diabetic patients, the results are often confusing [69,71,72,73]. Sotaniemi et al. [73] assessed the effects of diabetes on hepatic drug metabolism clearance in men using antipyrine in 298 diabetic patients, who were classified by type of the disease, age, gender, duration of therapy and liver involvement. They found that a 13-fold individual variation in antipyrine metabolism existed among all the diabetic patients. Antipyrine was eliminated faster in untreated type 1 patients, which was reversed by insulin treatment. Males aged 16–59 years, who responded insufficiently to insulin therapy, had a rapid antipyrine elimination, which could be normalized by the readjustment of insulin. The antipyrine elimination rate in women with insufficient glucose control on insulin therapy was comparable to controls. In type 2 diabetic patients, the clearance of antipyrine was unaltered in women, but men over 40 years of age showed a reduced antipyrine metabolism. Diet/drinking-habits also affect the expression of hepatic CYP450s. Urry et al. [74] investigated CYP1A2 activity in diabetic patients using coffee as a probe in 57 type 2 diabetes and 146 non-type 2 diabetes. They found that diabetes patients showed higher activity of CYP1A2 than control groups. Further studies showed that participants habitually consuming more caffeine showed higher CYP1A2 activity than participants with lower caffeine consumption. Several studies have demonstrated that CYP2D6 activity is unaltered by diabetes [43,69,75].

### 2.6. Interplay of Transporter-CYP450s in Liver

In the liver, drugs enter hepatocytes from portal blood through passive diffusion or transporter-mediated transport. In hepatocytes, drugs are metabolized by metabolic enzymes. Parent drugs and their metabolites are pumped out of hepatocytes to bile through efflux transporters (such as P-gp, BCRP or MRP2) and are exported to the blood through passive diffusion or efflux transporters (such as MRP3 and MRP4) (Figure 1B). Typical examples are statins. Most statins are substrates of CYP450s, MRPs, OATPs, BCRP and P-gp, characterizing transporter–enzyme interplay. The overall hepatic intrinsic clearance (CL_int,all_) is a hybrid parameter which is composed of intrinsic influx clearances (CL_int,up_) from portal blood, intrinsic metabolism clearance (CL_int,met_), biliary excretion clearance (CL_int,bile_) and intrinsic efflux clearance (CL_int,back_) of unbound drugs from hepatocytes to the blood, i.e.,
(1)$$CL_{\mathrm{int}t,all}=CL_{\mathrm{int},up}\times \frac{CL_{\mathrm{int},bile}+CL_{\mathrm{int},met}}{CL_{\mathrm{int},bile}+CL_{\mathrm{int},met}+CL_{\mathrm{int},back}}$$

According to the relative values of CL_int,bile_ and CL_int,met_*_,_* to CL_int,back_, the drugs are classified into three groups.

CL_int,back_ is much smaller than the sum of CL_int,bile_ and CL_int,met_, thus, CL_int,all_ is equal to CL_int,up_; this is to say, CL_int,all_ is only controlled by uptake clearance, which is mainly mediated by influx transporters. Typical drugs are statins.Sum of CL_int,bile_ and CL_int,met_ is much less than CL_int,back_. CL_int,all_ = CL_int,up_ × (CL_int,bile_ + CL_int,met_)/CL_int,back_, indicating that CL_int,all_ is determined by the net effect of CL_int,up_, CL_int,back_, CL_int,bile_ and CL_int,met_.Some drugs (such as midazolam) are not substrates of transporters. These drugs also rapidly penetrate the sinusoidal membrane, i.e., CL_int,up_
*=* CL_int,back_, thus, CL_int,all_ = CL_int,bile_ + CL_int,met_.

Diabetes significantly disorders transporter-CYP450 interplay via altering expressions and functions of transporters and CYP450s, finally affecting pharmacokinetics and activity/toxicity of drugs. For example, diabetes upregulated expressions and functions of hepatic CYP3A and OATP1B2, enhanced hepatic uptake and metabolism of simvastatin [12], atorvastatin [9,11], in turn, increasing hepatoxicity of atorvastatin [76]. Data from PBPK also showed that both hepatic OATP1B2 and CYP3A contribute to the clearance of atorvastatin, but the roles of hepatic OATP1B2 were much larger than that of CYP3A [11]. Another example is cyclosporin A. Cyclosporin A is also a substrate of CYP3A, MRP2, P-gp, OATP1B1 and BCRP. Significantly increased systemic clearance of cyclosporin A in diabetic rats [77] should be attributed to the net effect of alterations in these transporters and CYP3A. Methotrexate transport in the liver is mediated by various transporters such as OATP1B1, MRP2, BCRP and OAT2. Moreover, induction of CYP2E1 also enhanced methotrexate-induced hepatocytoxicity [78]. These results indicate that the regulation of hepatic OATPs and CYP2E1 expressions and functions may explain clinic findings that diabetic patients were particularly at increased risk of methotrexate hepatotoxicity [79].

## 3. Intestine

A series of transporters (such as P-gp, BCRP, OATPs, OCT1, PepTs, MRP2, MRP3 and MCTs) and enzymes (such as CYP450s and UGTs) have been expressed in enterocytes, implicating intestinal absorption of drugs and first-pass effects (Figure 2).

### 3.1. P-gp

P-gp is highly expressed at the apical membrane of the intestinal epithelium, pumping out of its substrates from enterocytes to the intestinal lumen. The expression of intestinal P-gp protein is region-dependent. P-gp protein progressively increases from proximal to distal regions. The highest expression of P-gp occurs at the ileum [80]. Several studies have demonstrated that diabetes impairs expression and function of intestinal P-gp, leading to the enhancement of intestinal absorption and increasing plasma exposure following oral administration of P-gp substrates such as arctigenin [81], protoberberine alkaloids [82], digoxin [83], grepafloxacin [84], paclitaxel [17] and morphine [85]. In line with the increased plasma levels of morphine, diabetic mice showed a stronger analgesic effect following an oral dose compared with control mice [85]. The decreased expression and function of intestinal P-gp under diabetes status may explain that clinic findings that diabetic patients also showed higher serum concentration of digoxin compared with non-diabetic patients following an oral dose of digoxin [86,87].

Downregulation of intestinal P-gp expression by diabetes was considered to be partly attributed to the acceleration of the ubiquitin-proteasome [88] via nitric oxide synthase (NOS) activation [89,90,91]. However, the role of NO in intestinal P-gp is time-dependent. In Caco-2 cells, it was reported that short-term exposure to sodium nitroprusside impaired P-gp function and expression, whereas long-term exposure stimulated P-gp function and expression [92]. Increased levels of short chain fatty acids (SCFAs) were also found in the intestinal content of diabetic rats [47]. A recent study showed that SCFAs downregulated the expression of intestinal P-gp via inhibiting histone deacetylase and NF-κB pathways [93].

### 3.2. MRP2

MRP2 (and MRP1) is co-located with phase II conjugating enzymes (such as UGTs and glutathione *S*-transferase) in the small intestine, pumping out of anions from enterocytes to the intestinal lumen. In contrast to P-gp, diabetes was reported to enhance the expression and function of intestinal MRP2 in rats [23]. Gliclazide is also a substrate of MRP2 [94]. C_max_ and AUC_0–tn_ of gliclazide following oral administration to diabetic rats were only 55% and 56% of those in control rats [94], respectively, which seemed to be attributed to the increased expression of intestinal MRP2. As expected, gliclazide administration significantly decreased glucose concentrations in healthy rats, but hypoglycemic effects of gliclazide were not observed in diabetic rats [95].

### 3.3. BCRP

Similarly to P-gp, BCRP is expressed at the luminal membrane of enterocytes. The BCRP protein gradually rises from proximal to distal regions. Diabetes-induced alterations in the expression of intestinal BCRP are dependent on the type of diabetes and duration of diabetes. STZ-induced diabetic rats showed significantly lowered expression and function of intestinal BCRP, accompanied by decreased intestinal clearance of glibenclamide and increased apparent intestinal effective permeability (P_eff_) [48], leading to the increased oral plasma exposure of glibenclamide following oral administration to diabetic rats. However, in STZ/HFD-induced diabetic rats, 10-day diabetic rats showed significantly lowered expression and function of intestinal BCRP, but remarkable inductions of intestinal BCRP expression and function were found in 22-day diabetic rats [11]. The induction of intestinal BCRP was considered to be related to the increased levels of short chain fatty acids (SCFAs) via activating PPARγ [96].

### 3.4. PepT1

PepT1, located on the luminal membrane of enterocytes, is responsible for intestinal absorption of peptidomimetic drugs, dipeptides and tripeptides resulting from the dietary breakdown of proteins and bacterial peptidomimetics. Several investigations have demonstrated that diabetes downregulated the expression and function of intestinal PepT1 in experimental animals [47,97,98,99], decreasing oral plasma exposures of cephalexin and acyclovir following oral dose of cephalexin and valacyclovir [99]. Importantly, altered expression and function of intestinal PepT1 are also dependent on sex. For example, in Sprague-Dawley rats, it was reported that STZ-induced diabetes decreased the expression and function of intestinal PepT1 in male rats, but increased them in female rats [100]. Leptin and insulin deficiencies may contribute to the downregulation of intestinal PepT1 [101,102]. Dyshomeostasis of bile acid compositions was observed in intestinal content of diabetic rats. Bile acids downregulated the expression of intestinal PepT1 via activating farnesoid X receptor (FXR) [99].

### 3.5. MCT6

MCT6, located on the apical side of human intestinal villous epithelial cells, mediates intestinal absorption of nateglinide [103]. Loop diuretics such as furosemide, piretanide, azosemide, and torasemide may also be substrates of MCT6 [104]. Diabetes significantly decreased the expression of intestinal MCT6 and decreased plasma exposure of nateglinide following oral dose to rats. Further study demonstrated that SCFAs, especially butyrate, downregulated the expression of intestinal MCT6 via activating PPARγ [47]. The downregulation of intestinal MCT6 may also partly explain that diabetes significantly decreased plasma exposures of furosemide [105] and azosemide [106] following oral doses to rats.

### 3.6. CYP450s

Intestine also highly expresses CYP450s, especially CYP3As, which mediate drug metabolism, contributing to first-pass effects. In STZ-induced diabetes, it was found that diabetes significantly decreased CYP3A activity. In line with this, formations of norverapamil [27] and 6-β hydroxyltestosterone [107] were reduced to 21% and 50% that of control rats, respectively. Significant downregulation of intestinal CYP3A expression and function was also found in STZ/HFD-induced diabetic rats [11] and in Tsumura Suzuki obese diabetic (TSOD) mice [108]. Insulin treatment partly reversed the decreased CYP3A expression in STZ-induced diabetic rats [107], but not in TSOD mice [108]. However, the real mechanism leading to the decreased expression of intestinal CYP3A was unclear. In generally, expressions of CYP3A and P-gp were mainly controlled by pregnane X receptor (PXR). We recently reported upregulation of intestinal FXR [83]. A report showed that FXR agonist GW4064 remarkably induced expression of small heterodimer partner (SHP) [109], in turn, inhibiting the transcriptional activity of PXR [109,110]. Moreover, levels of bile acids, FXR agonist, were increased in intestinal contents of diabetic rats, which inhibited the expression of intestinal CYP3A and P-gp via the activation of FXR-SHP pathway.

### 3.7. Transporter-CYP450 Interplay in Intestine

Some drugs are often substrates of various transporters and CYP3As. Alterations in oral plasma exposure of drug should be attributed to common effects of the altered transporters and CYP3As. For example, atorvastatin is a substrate of intestinal P-gp, BCRP and OATPs, indicating that the P_eff_ of atorvastatin should be the integrated effects of passive diffusion, OATP1A5-mediated absorption, P-gp mediated efflux and BCRP-mediated efflux. Diabetes also downregulated the expression of intestinal OATP1A5 [11]. Thus, the P_eff_ of atorvastatin under diabetic status should be the net effects of these altered transporters. Downregulation of intestinal CYP3A expression by diabetes was also involved in the first-pass effect of atorvastatin. PBPK simulation demonstrated that contributions of intestinal transporters and CYP3A was intestinal BCRP > intestinal CYP3A > intestinal P-gp > intestinal OATP1A5 [11]. Cyclosporin A is also a substrate of BCRP, OATPs, CYP3A and P-gp, indicating that increased oral plasma exposure of cyclosporin A in diabetic rats [111] was also partly attributed to the common effects of the intestinal transporters and CYP3A. Verapamil is a substrate of P-gp and CYP3As. PBPK simulation also showed that the contribution of intestinal P-gp was much larger than that of intestinal CYP3A [93], indicating that increased oral plasma exposure of verapamil [27] was mainly attributed to downregulation of intestinal P-gp. Grepafloxacin [112] is also a substrate. Decreased expression of BCRP may partly explain why the decreased secretory transport of grepafloxacin was not in line with the decrease in expression of P-gp protein in diabetic rats [84]. Similarly, intestinal OATP1A5 also mediates intestinal absorption of glibenclamide [113], inferring that the increased P_eff_ of glibenclamide by impairment of intestinal BCRP [48] may be partly weakened by the decreased expression of OATP1A5 protein.

## 4. Kidney

Drugs are eliminated in the kidney via urinary excretion. The process consists of glomerular filtration, secretion and reabsorption at the renal tubule. The secretion and reabsorption are mainly mediated by transporters. The identified renal transporters include OAT1/3, OCT1/2, PepT1/2, MATE1, MATE2/K, P-gp, BCRP, MCTs and urate transporter 1 (URAT1) (Figure 3A). Transporter interplay occurs at the kidney. For example, metformin, a substrate of OCTs and MATEs, is taken in renal epithelial cells via OCT1/2 at the basolateral membrane, then secreted into urine via MATEs at the brush-border membrane of renal tubular cells, mediating the renal excretion of metformin. The transporter interplay also participates in renal excretion of uric acid. URAT1, OAT1, OAT3, OAT4, OAT10, BCRP, sodium-dependent phosphate transport protein 1/4 (NPT1/4) and glucose transporter 9 (GLUT9) work together to regulate the excretion of uric acid via the kidney [114] (Figure 3B).

### 4.1. OAT1 and OAT3

The basolateral membrane of renal proximal tubule cells highly expresses OAT1 and OAT3, mediating the excretion of their substrates including ACE inhibitors, angiotensin II receptor blockers, diuretics, β-Lactam antibiotics, antiviral agents and endogenous compounds [3]. Several studies demonstrated that diabetes also impaired the function and expression of renal OATs. In STZ-induced diabetic rats, it was found that diabetes significantly decreased the membrane expression of renal OAT3, leading to lower uptake of [^3^H] estrone sulfate in renal cortical slice. These decreases were in line with the activation of PKCα and NF-κB pathways, increased nuclear factor erythroid 2-related factor 2 (Nrf2) and oxidative stress [115,116]. Alteration in OATs by diabetes is dependent on sex. In lean and obese Zucker spontaneously hypertensive fatty (ZSF1) rats, it was found that the levels of renal OAT1 and OAT3 mRNA were higher in lean females than in lean males. Obesity remarkably reduced the expression of renal OAT1 and OAT3 in female ZSF1 rats but not in male ZSF1 rats [117]. Clinical trials showed that expressions of renal OAT1 and OAT3 in patients with diabetic kidney disease were less than 50% of the normal levels, which were in line with decreased urinary excretion of some organic acid metabolites and increased plasma levels of organic acid metabolites [118].

Diuretics show their diuretic effect via affecting proximal tubular epithelial cells. Some diuretics (such as torasemid and furosemide) are also substrates of OATs [3]. Several studies have showed that OAT1 or OAT3 deficiency damage the natriuretic effects of furosemide and bendroflumethiazide [119,120], inferring that the decreased expression of OATs by diabetes may impair the diuretic effect of diuretics. In line with the deduction, alloxan-induced diabetes significantly increased the plasma exposure of furosemide and decreased recovery from urine following intravenous dose to rats [105]. Importantly, diuretic efficiency, natriuretic efficiency, kaluretic efficiency and chloruretic efficiency of furosemide in diabetic rats showed a trend to decrease. Similarly, the diuretic efficiency, natriuretic efficiency, kaluretic efficiency and chloruretic efficiency of torasemide were significantly impaired both in alloxan-induced diabetic rats and STZ-induced diabetes, although its plasma exposure was unaltered [50]. These findings also may explain the clinical finding that diabetic patients need higher furosemide doses [121]. Moreover, sodium-glucose cotransporter (SGLT2) inhibitor empagliflozin is a substrate of OAT3. A report showed that OAT3 deficiency damaged the glucosuric effect of empagliflozin [122], indicating that the decreased expression of renal OAT3 by diabetes may attenuate the glucosuric effect of empagliflozin, which may also explain the clinic finding that compared with normal renal function and normal-to-mildly reduced renal function, diabetic patients with mild-to-moderately reduced renal function showed the lowest lowering glucose effect of luseogliflozin [123].

### 4.2. OCTs

Renal OCT1/2, mainly expressed at the basolateral membrane of tubule cells, transport a variety of organic cations, such as metformin, cisplatin, cephalexin and acyclovir. In STZ/HFD-induced diabetic rats, it was found that OCT2 protein expression was only 50% of control rats [124]. Similarly, in STZ-induced diabetic rats, the expressions of renal OCT1 and OCT2 were decreased by 50% and 70%, respectively [125]. The reduced mRNA and protein expressions of OCT1, OCT2 and OCT3 under diabetic status were also associated with a reduction in the clearance of N^1^-methylnicotinamide [126,127]. The decreased mRNA and protein expressions of renal OCT2/3 were reported to be negatively correlated with the accumulation of renal and plasma advanced glycation end-products (AGEs) [127]. Insulin or AGE inhibitor aminoguanidine could reverse the decreased OCT2/3 by diabetes [125,127], inferring that the accumulation of AGEs may be involved in impaired expression of renal OCTs by diabetes [127]. Expressions of OCT1 and OCT2 are also be dependent on sex. In lean female ZSF1 rats, the mRNA expression level of OCT1 was significantly higher than that in their male counterparts. Obesity significantly reduced renal mRNA expressions of OCT1 and OCT2 in female ZSF1 rats, but not in male rats [117]. The decreased expression of renal OCTs by diabetes may affect pharmacokinetics of their substrates. For example, significantly higher AUC of metformin and lower renal clearance were observed in alloxan-induced diabetic rats than in normal rats following an intravenous dose [128]. A clinic trial [129] showed that renal clearance of metformin decreased along with diabetes progression. A report demonstrated the rank of the renal clearances of metformin in healthy subjects (525 ± 125 mL/min) > newly diagnosed maturity onset diabetic patients (322 ± 166 mL/min) > maturity onset diabetic patients (224 ± 38 mL/min). Interestingly, pregnancy seems to increase the functions of renal MATEs and OCTs [130]. Renal clearances of *N*^1^-methylnicotinamide (endogenous probe for the renal OCTs and MATEs) in both mid (504 ± 293 mL/min) and late pregnancy (557 ± 305 mL/min) were reported to be higher than in postpartum (240 ± 106 mL/min). The renal secretion of *N*^1^-methylnicotinamide was 3.5-fold higher in mid pregnancy and 4.5-fold higher in late pregnancy compared with postpartum [130]. In line, gestational diabetes mellitus pregnant women showed higher renal clearance and renal secretion [131], and lower plasma concentrations of metformin [131,132] compared with non-pregnant controls. Another example is cisplatin. Cisplatin is a substrate of OCTs. Its main toxicity is nephrotoxicity. Several studies have demonstrated that both insulin-dependent and insulin-independent diabetes show resistance against cisplatin-induced nephrotoxicity [133,134,135]. Animal experiments showed that diabetes prevented nephrotoxicity partly via decreasing the renal accumulation of cisplatin [136,137]. However, cisplatin treatment was not beneficial in diabetes due to its compromising its antitumor effect [136].

### 4.3. Other Transporters

Diabetes induced an approximately two-fold increase in renal PepT1 protein and slightly induced mRNA of PepT1 and PepT2 [138]. Reports on the expression of ABC transporters are often contradictory. For renal P-gp, it was reported that spontaneous type 1 and type 2 diabetic mice showed lower expressions of renal P-gp protein. The high glucose was considered to be a reason reducing the expression and function of P-gp [139]. However, the expression of P-gp protein was also reported to be unchanged [17,111,138], although level of Mdr1a mRNA was increased [17,138].

For MRP2, increased expression of renal MRP2 protein was found in STZ/HFD-diabetic rats [124] and in STZ-induced diabetic rats [23]. For BCRP, HFD/STZ-induced diabetes significantly increased the expression of BCRP protein in rats, but STZ-induced diabetes significantly decreased mRNA level of renal BCRP in rats [136]. These discrepancies may result from the duration of diabetes, sex and the type of diabetes. For example, in female rats, obesity decreased the expressions of MRP4 and Mdr1b mRNA, but the expression of MRP2 mRNA was unaltered. In male ZSF1 rats, expressions of MRP2, MRP4 and Mdr1b mRNA remained unchanged in obese rats compared to lean rats [117].

### 4.4. CYP450

The identified renal CYP450s include CYP2C, CYP2J, CYP2E1, CYP4As and CYP4Fs. They mainly mediate local biotransformation of endogenous compounds such as arachidonic acid. For example, arachidonic acid is metabolized by CYP4As and CYP4Fs to 20-hydroxyeicosatetraenoic acid (vasoconstrictor) and by CYP2C and CYP2J to epoxyeicosatrienoic acid (vasodilator), synergistically regulating renal vasoactivity. Several reports have shown that diabetes also affect the expressions of renal CYP450s. In mice, it was found that HFD feeding significantly decreased activities of CYP3A (1′-hydroxymidazolam), CYP2E1 (hydroxychlorzoxazone), CYP2J (ebastine hydroxylation) and CYP2B6 (hydroxybupropion) and CYP4A (12-hydroxydecanoic acid), but increased the activities of CYP2C (hydroxytolbutamide) and CYP2D (Hydroxybufuralol) [140]. Diabetes affects the expressions and activities of renal CYP450s in an isoform- and species-specific manner. In STZ-induced diabetic rats, formations of hydroxyandrostenedione and 2α-hydroxytestosterone were significantly increased by 250% and 300% compared to control rats, but formations of androstenedione and 6β-hydroxytestosterone were significantly decreased. The formations of 16α-hydroxytestosterone and 7α-hydroxytestosterone were unaltered [141]. Induction of CYP2E1, 4A2 and K-4 as well as increases in omega- and (omega-1) hydroxylation of lauric acid were also observed in kidney of STZ-induced rats [26]. Insulin treatment partly reversed these alterations by diabetes [26,141]. However, a recent report showed that expression of CYP4A protein in diabetic mice induced by HFD/STZ was downregulated by 29.16% of control mice [142]. The roles of altered renal CYP450s by diabetes in disposition of drugs needed further investigation.

### 4.5. Transporter Interplay in Kidney

Transporter interplay occurs in the kidney. For example, methotrexate is eliminated mainly via renal active excretion, which is involved in OAT1/3 at basolateral membrane of tubule and BCRP and MRP2 at brush-border membrane of tubule, these transporters in series work to regulate renal active excretion of methotrexate. Both higher plasma concentration of methotrexate and higher toxicity of methotrexate in STZ-diabetic rats [143] may partly be explained by the imbalance of expressions of renal OATs, BCRP and MRP2. Another example is β-lactam antibiotics. OATs mediate the uptake of β-lactam antibiotics into the tubule from the blood and PepT1/2 located in the brush-border membrane of the tubule, mediate reabsorption of antibiotics from the urine, and regulate urinary excretion of β-lactam antibiotics. Downregulation of renal OATs may partly contribute to a decrease in the renal clearance of DA-1131 (a carbapenem antibiotics) [144] and lower renal cortical cephaloridine accumulation in diabetic rats [145]. Creatinine is also substrate of OCT1/2, MATEs and OAT1/3, indicating that the decreased expressions of renal OATs and OCTs at least partly contribute to decreased clearance of creatinine under diabetic status.

## 5. Application of PBPK to Transporter-Enzyme Interplay

The above statements demonstrated that diabetes may simultaneously affect functions and expressions of transporters and enzymes in the intestine, liver and kidney. Their contributions to the disposition of drugs are often opposite. Moreover, some physiological parameters such as blood flow rates, intestinal transit and the free fraction of drug in plasma are often altered. Thus, the effects of diabetes on pharmacokinetics of drugs should be the integrated effects of these alterations, which may be accomplished using PBPK.

### 5.1. Atorvastatin

Atorvastatin is a substrate of P-gp, OATPs, BCRP and CYP3As. Diabetes reduced expressions of intestinal P-gp, CYP3A and OATP1A5, while induced expressions of liver OATP1B2 and CYP3A. Intestinal BCRP was highly dependent upon diabetes course, which was decreased at an early phase of diabetes (10-day) but induced at a late phase (22-day). A pharmacokinetic study showed that oral plasma exposure was increased in 10-day diabetic rats, but decreased in 22-day diabetic rats. PBPK simulation demonstrated that contributions of these targeted protein to altered oral plasma exposure of atorvastatin were intestinal BCRP > hepatic OATP1B2 > hepatic CYP3A > intestinal CYP3A > intestinal P-gp > intestinal OATP1A5. The increased oral plasma atorvastatin exposure in 10-day diabetic rats and the decreased oral plasma atorvastatin exposure in 22-day diabetic rats were attributed to the altered intestinal BCRP [11].

### 5.2. Verapamil

Verapamil is substrate of CYP3As and P-gp. Rat experiments showed that diabetes increased plasma exposure of verapamil following an oral dose but decreased plasma exposure of verapamil following an intravenous dose [29]. A semi-PBPK model (Figure 4) was successfully developed to simulate the pharmacokinetics of verapamil (Figure 5A,B) in rats using the parameters listed in Table 1 and Table 2. Intestinal P-gp in diabetic rats was set to 60% of control rats [82]. The contributions of intestinal/hepatic CYP3A and intestinal P-gp to alterations in oral plasma concentration of verapamil in diabetic rats were investigated. The results demonstrate that increased oral plasma exposure of verapamil was mainly attributed to the impairment of intestinal P-gp (Figure 5E) and the role of intestinal CYP3As was minor (Figure 5D). Furthermore, the contribution of increased expression of hepatic CYP3As to the altered plasma concentration of verapamil following oral dose to rats was less than that of intestinal P-gp impairment, whose net effect was to increase oral plasma exposure of verapamil (Figure 5G).

### 5.3. Furosemide

Rat experiments demonstrated that diabetes decreased plasma furosemide exposure following oral dose but increased plasma furosemide exposure following intravenous dose [105]. Intestinal absorption of furosemide may be involved in MCT6 [84]. It was assumed that intestinal absorption of furosemide was mediated by MCT6. Function of intestinal MCT6 under diabetic status was set to be 50% according previous report [47]. Furosemide is eliminated mainly via renal secretion due to OAT1/2 [3]. Part of furosemide is metabolized by CYP2C11, CYP3As and CYP2E1 [152]. The contribution of CYP2C11 was 61.5% of the total metabolism [152], rests (38.5%) were assumed to attributed to CYP2E1 and CYP3As. Hepatic clearance (CL_liver_) of furosemide was reported to be 2.20 mL/min/kg [105] and the estimated value of fu × CL_int,liver_ was 0.75 mL/min/250g. Diabetes increased the expression of CYP2E1 and CYP3As by 3-fold [28] and the expression of CYP2C11 mRNA was decreased to 16% that of control rats [47]. Thus, fu × CL_int,liver_ and CL_liver_ were estimated to be 0.75 mL/min/250g and 2.20 mL/min/kg. The estimated CL_liver_ of furosemide was near to observed data (2.98) in diabetic rats [105]. Diabetes altered expression of these targeted proteins (CYP2C11, CYP3As, CYP2E1 and OATs), intestinal blood, liver blood flow and renal blood flow. It was assumed that intestinal absorption of furosemide was mediated by MCT6. All these contribute to the altered furosemide pharmacokinetics, whose integrated effects (Figure 6) were predicted using PBPK model. The results demonstrate that the contribution of increases in CYP3A and CYP2E1 to hepatic clearance of furosemide was partly attenuated by decreases in CYP2C11, which may explain why non-renal clearance of furosemide was only slightly increased under diabetic status [105]. PBPK simulation demonstrated that decreases in renal excretion of furosemide due to downregulation of renal OATs and renal blood flow rates mainly contributed the increased plasma exposure of furosemide following an intravenous dose. Decreased oral exposure of furosemide was mainly attributed to the impairment of intestinal absorption due to the suppression of intestinal MCT6. The contribution of decreased expression of renal OATs to the altered oral plasma exposure of furosemide was less than that of intestinal MCT6 impairment, whose net effect was to decrease oral plasma of furosemide (Figure 6G).

### 5.4. Metformin and Nisoldipine 

A whole-body PBPK model was also used to predict the pharmacokinetics of metformin in diabetic patients [153]. The results show that increased plasma levels of metformin following oral dose mainly resulted from both the impairment of renal function and delaying gastrointestinal transit. The metabolism of nisoldipine is mediated by CYP3As. Decreased CYP3A activity may partly explain the increased oral plasma exposure. However, when nisoldipine, in controlled-release tablets, was given to diabetic patients, the roles of gastrointestinal transit were nonnegligible. PBPK simulation also demonstrated that delaying gastrointestinal transit increased the intestinal absorption of nisoldipine, indicating that increased oral plasma of nisoldipine following oral administration of nisoldipine controlled-release tablet in diabetes patients was mainly attributed to both decreased CYP3As and delaying gastrointestinal transit.

## 6. Future Perspective

Growing evidence has demonstrated that diabetes often simultaneously affects transporters and enzymes in the intestine, liver and kidney. Moreover, these alterations are dependent on tissue and diabetic progression. For example, diabetes induced the expression of hepatic CYP3A, but downregulated the expression of intestinal CYP3A. In STZ/HFD-induced diabetic rats, 10-day diabetes decreased the expression of both intestinal and hepatic BCRP, but 22-day diabetes induced the expression of intestinal BCRP not hepatic BCRP protein [11]. Moreover, mRNA levels are often indexed as the expression of targets, but mRNA levels do not always reflect the protein expression and activity. For example, a report showed that diabetes induced a five-fold increase in the expression of renal Mdr1a mRNA in rats, but the expression of renal P-gp protein was unaltered [138]. Similarly, mRNA expression of BCRP in the liver of GK rats was increased by 20-fold, but the expression of BCRP protein and function (biliary excretion of rosuvastatin) was unchanged [19]. In contrast, diabetes decreased the level of hepatic MRP2 protein by 80% in rats without affecting the mRNA expression of MRP2 [22]. Similarly, no correlation between the mRNA level and activity (chlorzoxazone 6-hydroxylation activity) of CYP2E1 was also found in human liver [154]. All these indicate that data based on mRNA sometimes give the wrong conclusion. Clinical trials [66,68,73,74,131,132] have also demonstrated that many factors such as type of diabetes, diabetic progression, gender, age, therapy/effectiveness, diet/drinking habits, complications and co-medicines affect expressions and functions of CYP450s and transporters, which may partly explain contradictory effects of diabetes on expressions of CYP450/transporters and pharmacokinetics. Alterations in expressions and functions of transporters and CYP450s by diabetes are dependent on specific isoforms (CYP450s and transporters) and tissues. Moreover, physiological parameters such as blood flow rates in tissues are altered. All these results indicate that alterations in the disposition of drugs under diabetes should be attributed to the integrated effect of these factors. We thought that PBPK model could be used to accomplish the prediction of drug disposition under diabetic status. Furthermore, the PBPK model could separately illustrate individual contributions of each factor to drug disposition and their integrated effects.

## Figures and Tables

**Figure 1 pharmaceutics-12-00348-f001:**
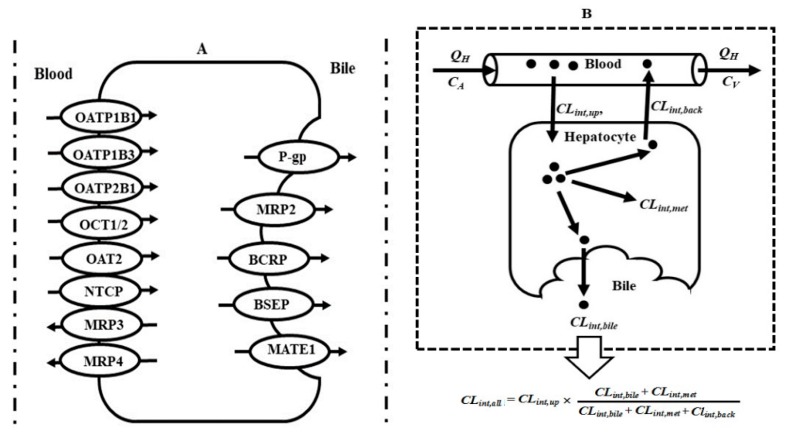
(**A**) Possible location of main transporters in liver. (**B**) Transporter–enzyme interplay in the elimination of drugs in the liver. Symbol: BCRP, breast cancer resistance protein; BSEP, bile salt export pump; C_A_, concentrations of drug in arterial blood; CL_int,all_, overall hepatic intrinsic clearance; CL_int,up_, intrinsic uptake clearance; CL_int,back_, intrinsic clearance of backflux to blood; CL_int,met_, intrinsic metabolism clearances; CL_int,bile_, biliary clearance of unbound drug; C_V_, concentrations of drug in venous blood; MATEs, multidrug/toxin extrusion; MRPs, multidrug resistance-associated proteins; NTCP, sodium taurocholate co-transporting polypeptide; OATs, organic anion transporters; OATPs, organic anion transporting polypeptides; OCTs, organic cation transporters; Q_H_, hepatic blood flow; P-gp, P-glycoprotein.

**Figure 2 pharmaceutics-12-00348-f002:**
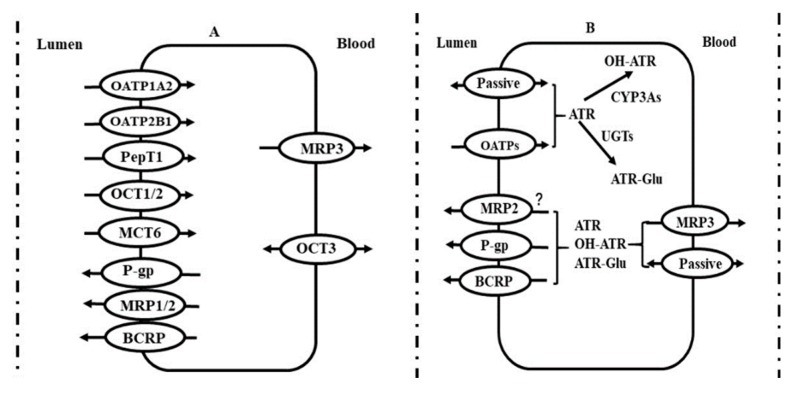
(**A**) Possible location of main transporters in human intestine and (**B**) roles of transporter-CYP3A interplay in disposition of atorvastatin in enterocytes. Symbol: ATR, atorvastatin; BCRP, breast cancer resistance protein; MCTs, monocarboxylate transporters; MRPs, multidrug resistance-associated proteins; OATPs, organic anion transporting polypeptides; OCTs, organic cation transporters; PepT1, peptide transporters; P-gp, P-glycoprotein; OH-ATR, hydroxyl atorvastatin; ATR-Glu, atorvastatin acyl glucuronide.

**Figure 3 pharmaceutics-12-00348-f003:**
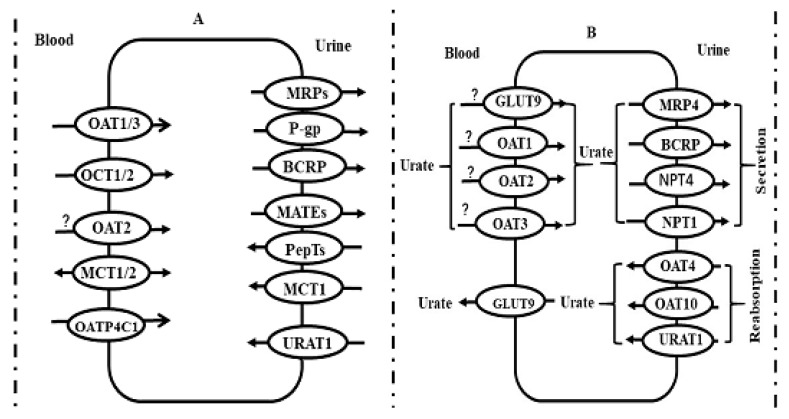
(**A**) Possible location of main transporters in human kidney and (**B**) roles of transporter interplay in renal excretion of uric acid. Symbol: OATPs, organic anion transporting polypeptides; OATs, organic anion transporters; OCTs, organic cation transporters; PepTs, peptide transporters; P-gp, P-glycoprotein; MRPs, multidrug resistance-associated proteins; BCRP, breast cancer resistance protein; MATEs, multidrug and toxin extrusion, NPTs, sodium-dependent phosphate transport proteins, GLUT9, glucose transporter 9, URAT1, urate transporter 1.

**Figure 4 pharmaceutics-12-00348-f004:**
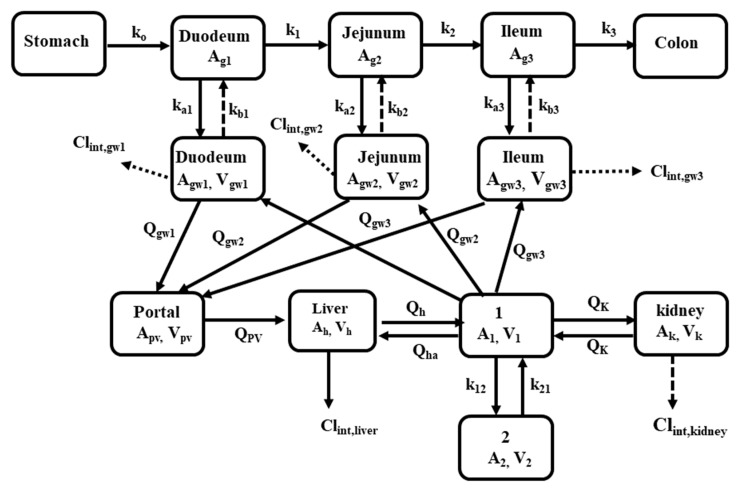
Schematic diagram of semi-physiologically based pharmacokinetic model (PBPK) model describing the pharmacokinetics of verapamil and furosemide in rats. A_i_, Q_i_ and V_i_ indicate drug amount, blood flow and volume in corresponding compartment, respectively. k_i_, k_a,i_ and k_b,i_ represent the transit rate constant, drug absorption rate constant and efflux rate constant from enterocytes to the gut lumen, respectively. Cl_int,gwi_, Cl_int,liver_ and Cl_int,kidney_ mean the intrinsic clearance in enterocytes, hepatocytes and kidney, respectively.

**Figure 5 pharmaceutics-12-00348-f005:**
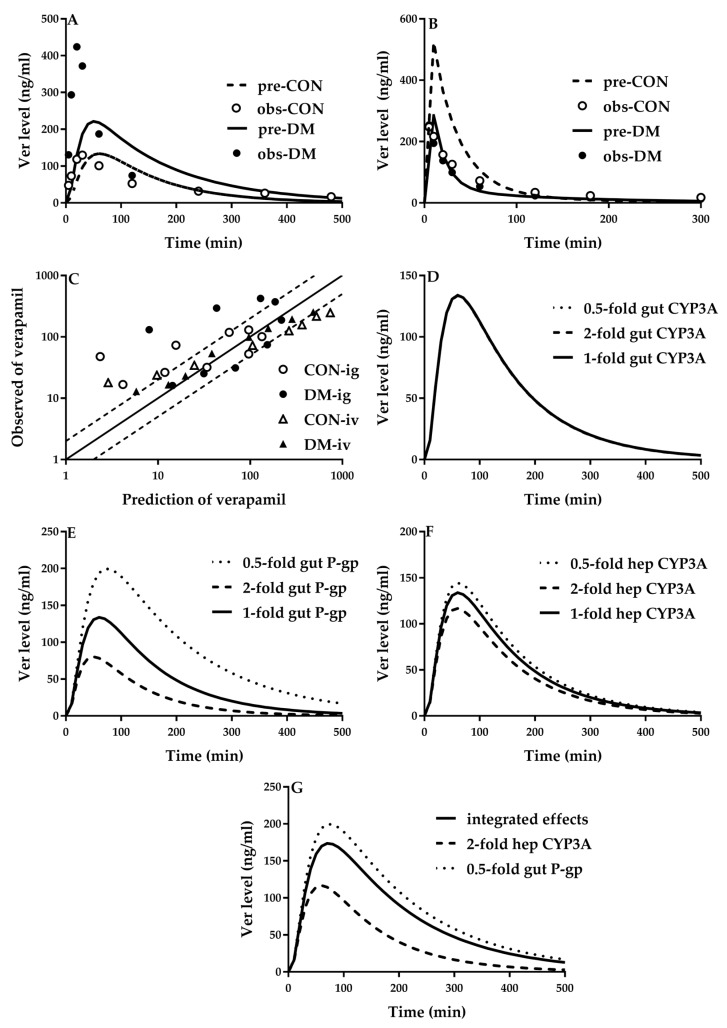
Observed (point) and predicted (line) plasma concentrations of verapamil (Ver) following (**A**) oral (10 mg/kg, ig) and (**B**) intravenous dose (1 mg/kg, iv) to diabetic rats (DM) and control rats (CON); (**C**) comparison of mean observed and predicted plasma concentrations of verapamil at each time point; (**D**) individual contributions of altered hepatic CYP3A, (**E**) intestinal CYP3A and (**F**) intestinal P-gp to oral plasma exposure of verapamil as well as (**G**) their integrated effects. The observations were obtained from the literature [27].

**Figure 6 pharmaceutics-12-00348-f006:**
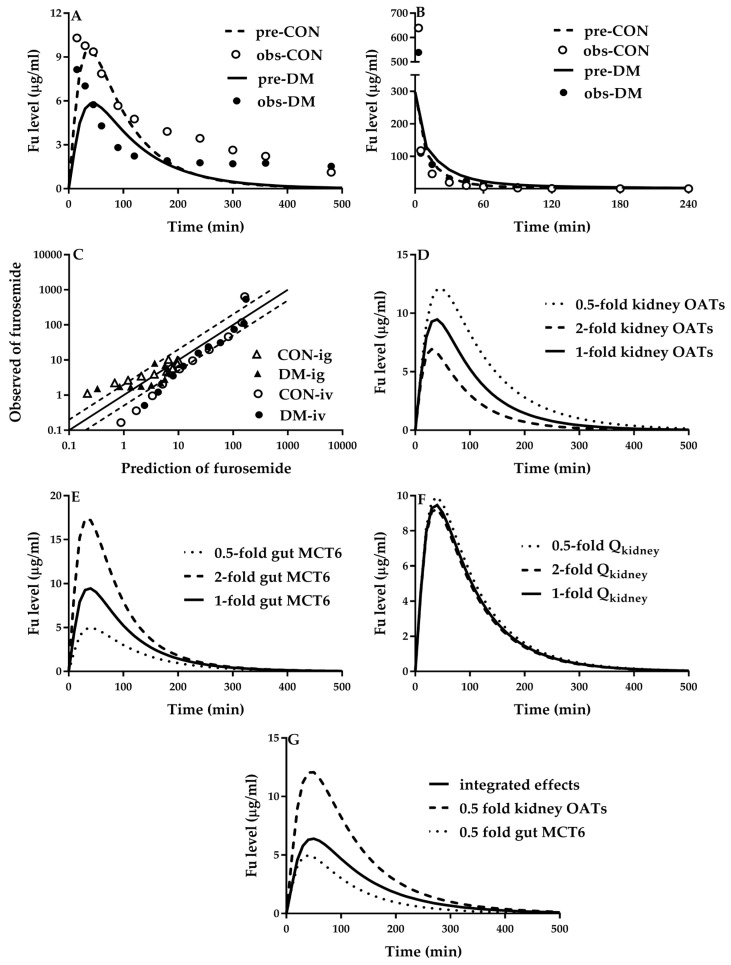
Observed (point) and predicted (line) plasma concentrations of furosemide (Fu) following (**A**) oral (6 mg/rat, ig) and (**B**) intravenous dose (6 mg/rat, iv) to diabetic rats (DM) and control rats (CON). Comparison of mean observed and predicted plasma concentrations of furosemide at each time point (**C**). Contributions of altered (**D**) renal OATs, (**E**) intestinal MCT6 and (**F**) renal blood flow (Q_kidney_) to oral plasma exposure of furosemide as well as (**G**) integrated effects of intestinal MCT6 impairment and renal OATs impairment. The observations were obtained from the literature [89].

**Table 1 pharmaceutics-12-00348-t001:** The physiological parameters of rats for the semi-PBPK model [11].

Parameters	Unit	Control Rats	Diabetic Rats
Gastric emptying rate	h^−1^	20.8	20.8
Duodenum transit time	h^−1^	28.74	28.74
Jejunum transit time	h^−1^	4.2	4.2
Ileum transit time	h^−1^	0.789	0.789
Intestinal radius	cm	0.2	0.2
Duodenum wall volume	mL	1.08	1.08
Jejunum wall volume	mL	9.94	9.94
Ileum wall volume	mL	0.32	0.32
Portal vein volume	mL	0.25	0.25
Liver volume	mL	10	10
Renal volume	mL	1.83 [146]	1.83
Duodenum wall blood flow	mL/min	0.972	2.223
Jejunum wall blood flow	mL/min	9.125	20.877
Ileum wall blood flow	mL/min	0.253	0.580
Portal vein blood flow	mL/min	16.043	23.68
Hepatic artery blood flow	mL/min	2.243	11.914
Liver blood flow	mL/min	18.286	35.594
Renal blood flow	ml/min	11.7 [146]	4.10^a^
Hepatic microsomal protein	mg/g liver	44.8	44.8
Intestinal microsomal protein	mg/g intestine	25.9	25.9
Liver weight	g/kg body weight	40	36

^a^ Renal blood flow was reduced by 65% of control rats [147].

**Table 2 pharmaceutics-12-00348-t002:** Pharmacokinetic parameters of verapamil and furosemide for PBPK simulation in diabetic rats (DM) and control rats (CON).

Parameter	Unit	Furosemide		Verapamil	
		CON rats	DM rats	CON rats	DM rats
Vc	L/kg	0.127 [148]	0.127	0.505 ^a^	0.505
k21	h^−1^	0.835 [148]	0.835	11.880 ^a^	11.88
k12	h^−1^	0.989 [148]	0.989	10.740 ^a^	10.74
fu	%	10.4 [105]	10.4 [89]	0.05 [93]	0.05
Kt:p	Liver	0.33 [148]	0.33	8.20 ^b^	8.20
Kt:p	Intestine	0.517 [148]	0.517	319.39 ^b^	319.39
Kt:p	Kidney	1.36 [148]	1.36	/	/
P_app,A-B_ (caco-2)	cm/s × 10^−6^	6.90 [149]	3.45 ^c^	13.8	13.8
P_app,B-A_ (caco-2)	cm/s × 10^−6^	/	/	24.84	14.90 ^d^
CL_kidney_	mL/min/kg	4.33 [105]	/		
CL_liver_	mL/min/kg	2.20 [105]	/		
Fu × CL_int,liver_	mL/min/250 g	0.60 ^e^	0.75 ^f^	/	/
Fu × CL_int,kidney_	mL/min/250 g	1.19 ^e^	0.48 ^g^	/	/
Microsomes					
Liver					
V_max_	nmol/(min/mg prot)	/	/	1.60 [27]	2.38 [27]
K_m_	μM	/	/	13.21 [27]	16.09 [27]
Intestine					
V_max_	pmol/(min/mg prot)	/	/	49.04 [27]	22.70 [27]
K_m_	μM	/	/	34.06 [27]	55.37 [27]

^a^ Estimated using the reported data [150]; ^b^ estimated according to a method [151] reported by Ruark et al. and physicochemical properties of verapamil; ^c^ function of intestinal MCT6 was set to be 50% that of control rats [47]; ^d^ level of intestinal P-gp protein was 60% that of control rats [82]; ^e^ fu × Clint were estimated using equation fu*Clint = Q × CL/(Q − CL); ^f^ contribution of CYP2C11 was 61.5% of the total metabolism [152], rests (38.5%) were assumed to attributed to CYP2E1 and CYP3As. Diabetes increased the expression of CYP2E1 and CYP3As by 3-fold [28] and expression of CYP2C11 mRNA was decreased to 16% that of control rats [47]; ^g^ function of renal OATs was decreased to 40% that of control rats [115].

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
