# Peer review of "Imbalance of Drug Transporter-CYP450s Interplay by Diabetes and Its Clinical Significance"

_pharmaceutics, 2020, doi:10.3390/pharmaceutics12040348_

Round 1

Reviewer 1 Report

I have no major concerns about the scientific content of this manuscript. However, the English used is requiring extensive editing, as it is highly rebutant and annoying for a reviewer in the present form. And so would it be for the potential readers. Figures are not user-friendly are hard to read/understand, especially fig 5-6. These are major issues as this paper is presented as a review, which should be crystal clear.

For instance, these are changes I would make only in the abstract to make it more readable. 

Abstract: Pharmacokinetics of a drug is highly dependent on coordinate work of influx transporters, enzymes and efflux transporters (i.e. transporter-enzyme interplay), which occurs in various tissues including liver, kidney and intestine. The identified drug-related influx transporters include organic anion transporting polypeptides (OATPs), organic anion transporters (OATs), organic cation transporters (OCTs), peptide transporters (PepTs) and monocarboxylate transporters (MCTs). The identified efflux transporters related to drugs are P-glycoprotein (P-gp), multidrug resistance-associated proteins (MRPs), breast cancer resistance protein (BCRP) and multidrug/toxin extrusions (MATEs). Enzymes related to drug metabolism are mainly cytochrome P450 enzymes (CYP450s) and UDP-glucuronosyltransferases (UGTs). Accumulating evidence has demonstrated that diabetes alters expression and function of these transporters and CYP450s in a different manner, disordering the transporter-enzyme interplay, and in turn, affecting pharmacokinetics of some drugs. We aimed to focus on (1) imbalance of transporter-CYP450 interplay in liver, intestine and kidney due to alterations in expression of influx transporters (OATPs, OCTs, OATs, PepTs and MCT6), efflux transporters (P-gp, BCRP and MRP2) and CYP450s (CYP3As, CYP1A2, CYP2E1 and CYP2Cs) under diabetic status; (2) the net contributions of these alterations in expression and function of transporters and CYP450s to drug disposition, therapeutic efficacy and drug toxicity; (3) application of a physiologically-based pharmacokinetic model in transporter-enzyme interplay.

Author Response

Point: the English used is requiring extensive editing, as it is highly rebutant and annoying for a reviewer in the present form. And so would it be for the potential readers. Figures are not user-friendly are hard to read/understand, especially fig 5-6. These are major issues as this paper is presented as a review, which should be crystal clear.

Response:   

Thank your advice, we have rewritten the revised manuscript in abstract. Fig 5 and 6 were rearranged.

Reviewer 2 Report

This review described the effect of diabetes on the drug transporter and CYP450s in animals and the humans. There are some comments

  1. This review described the more animal data than clinic data, and therefore, the title would be changed including the animal data.
  2. Section 4, CYPs in the kidney was not discussed. Why?
  3. Fig. 5 and 6: The references for data sources of PBPK simulation results of verapamil and furosemide should be described in the legends.
  4. Among the references cited in this review, the references published in the recent 5 years are 28 among 133 references. The recent papers and clinical papers would be added.
  5. line 51, ‘co-transporting polypeptide (NTCP)’ should be corrected to ‘sodium taurocholate co-transporting polypeptide (NTCP)’.
  6. Lines 57-60: The references for the sentence would be given.
  7. Line 96, The full names of PKCa and NF-kB should be given.
  8. Line 130, ~ AND ~ should be changed to ‘~ and ~”.
  9. Line 151, It is necessary to check that diclofenac acid should be corrected to diclofenac.
  10. Line 320, It is necessary to check either valacyclovir or acyclovir is correct. Reference 83 can’t be available on-line.
  11. Line 375, 383: The bold character would be changed to plain.
  12. Line 422, The bracket should be deleted.

Author Response

Point 1: This review described the more animal data than clinic data, and therefore, the title would be changed including the animal data.

Response 1:

Thank your advice. As you mentioned, although most of data resulted from animals, clinic data have been demonstrated in the reviews. Thus, title “Imbalance of drug transporter-CYP450s interplay by diabetes and its clinical significance” would still reflect findings in both animals and humans.   

Point 2: Section 4, CYPs in the kidney was not discussed. Why?

Response 2:

Thank your suggestion. We added renal CYP450s in the revised manuscript as follows.

“4.4 CYP450

The identified renal CYP450s include CYP2C, CYP2J, CYP2E1, CYP4As and CYP4Fs. They mainly mediate local biotransformation of endogenous compounds such as arachidonic acid. For example, arachidonic acid is metabolized by CYP4As and CYP4Fs to 20-hydroxyeicosatetraenoic acid (vasoconstrictor) and by CYP2C and CYP2J to epoxyeicosatrienoic acid (vasodilator), synergistically regulating renal vasoactivity. Several reports have showed that diabetes also affect expressions of renal CYP450s. In mice, it was found that HFD feeding significantly decreased activities of CYP3A (1′-hydroxymidazolam), CYP2E1 (hydroxychlorzoxazone), CYP2J (ebastine hydroxylation) and CYP2B6 (hydroxybupropion) and CYP4A (12-hydroxydecanoic acid), but increased activities of CYP2C (hydroxytolbutamide) and CYP2D (Hydroxybufuralol) [140]. Diabetes affects expression and activities of renal CYP450s in an isoform-specific and species manner. In STZ-induced diabetes rats, formations of hydroxyandrostenedione and 2a-hydroxytestosterone were significantly increased by 250% and 300% of control rats, but formations of androstenedione and 6b-hydroxytestosterone were significantly decreased. The formations of 16a-hydroxytestosterone and 7a-hydroxytestosterone were unaltered [141]. Induction of CYP2E1, 4A2 and K-4 as well as increases in omega- and (omega-1) hydroxylation of lauric acid were also observed in kidney of STZ-induced rats [26]. Insulin treatment partly reversed these alterations by diabetes [26,141]. However, a recent report showed that expression of CYP4A protein in diabetes mice induced by HFD/STZ was down-regulated by 29.16% of control mice [142]. The roles of altered renal CYP450s by diabetes in disposition of drugs needed further investigation.”

Point 3: Fig. 5 and 6: The references for data sources of PBPK simulation results of verapamil and furosemide should be described in the legends.

Response 3:

Thank your suggestion. We have added source of verapamil and furosemide in their legends.

In Fig 5. The observations were obtained from the literature [27].

In Fig 6. The observations were obtained from the literature [89].

Point 4: Among the references cited in this review, the references published in the recent 5 years are 28 among 133 references. The recent papers and clinical papers would be added.

Response 4:

Thank your suggestion. We have looked into publications on Pubmed. According to your suggestion, we added new references.

Point 5: line 51, ‘co-transporting polypeptide (NTCP)’ should be corrected to ‘sodium taurocholate co-transporting polypeptide (NTCP)’.

Response 5:

Thank your suggestion. We have corrected it in the revised manuscript.

Point 6: Lines 57-60: The references for the sentence would be given.

Response 6:

Thank your suggestion. We have added references in the revised manuscript.

Point 7: Line 96, The full names of PKCα and NF-kB should be given.

Response 7:  

Thank your suggestion. We have rewritten them in the revised manuscript.

Point 8: Line 130, ~ AND ~ should be changed to ‘~ and ~”.

Response 8:

Thank your suggestion. We have corrected it in the revised manuscript.

Point 9: Line 151, It is necessary to check that diclofenac acid should be corrected to diclofenac.

Response 9:

Thank your suggestion. We have corrected it in the revised manuscript.

Point 10: Line 320, It is necessary to check either valacyclovir or acyclovir is correct. Reference 83 can’t be available on-line.

Response 10:

Thank your suggestion. Valacyclovir is a prodrug of acyclovir, which can be converted into acyclovir in the body. The reference has been accepted for publication and will be online in the future.

Point 11: Line 375, 383: The bold character would be changed to plain.

Response 11:

Thank your suggestion. We have corrected it in the revised manuscript.

Point 12: Line 422, The bracket should be deleted.

Response 12:

Thank your suggestion. We have corrected it in the revised manuscript.

Reviewer 3 Report

In this manuscript, Liu and Yang thoroughly describe how diabetic status can modify expression of and, therefore, interplay between transporters and enzymes involved in drug metabolism in various human organs.

The language is the main concern at this stage of the manuscript. These are just a few examples: lines 15-16 in the abstract, lines 42-43 and 48-49 in the introduction, lines 283-285, lines 448-449 and so on. Therefore, extensive editing was recommended.

Here are several additional suggestions to improve this review article.

1) A list or table of abbreviations could be useful

2) Avoid jargon, for example, instead of "Western", use "immunoblotting" term

3) In this version, Figure 6 is too small and the font is unreadable

 I am looking forward to see a revised version of this manuscript.

Author Response

Point: The language is the main concern at this stage of the manuscript. These are just a few examples: lines 15-16 in the abstract, lines 42-43 and 48-49 in the introduction, lines 283-285, lines 448-449 and so on. Therefore, extensive editing was recommended.

Response:

Thank your suggestion, we have edited them in the revised manuscript. Here are several additional suggestions to improve this review article.

Point 1: A list or table of abbreviations could be useful.

Response 1:

Thank your suggestion, we have added list of abbreviations.

Point 2: Avoid jargon, for example, instead of "Western", use "immunoblotting" term

Response 2:  

Thank your suggestion, we have corrected it in the revised manuscript.

Point 3: In this version, Figure 6 is too small and the font is unreadable.

Response 3.

Thank your suggestion, we have redrawn Fig 6 in the revised manuscript.

Reviewer 4 Report

The current work summarized a relevant clinical pharmacological topic. The authors should be commended for their effort. 

The reviewer has the following comments which if addressed would improve the quality of this manuscript further. 

Major comments: 

  1. The manuscript provides sufficient references to relevant articles. But what is missing is the author's view on the cited manuscripts. It would be nice if the authors develop this topic around a central hypothesis and leading towards the requirement of PBPK model for the analysis/ understanding the effect of diabetes on CYP-transporter interplay. 
  2. What mechanistic understanding can PBPK provide over the use of PopPK model where "diabetes" would be used as a categorical covariate? Can the current framework of PBPK model provide a mechanistic understanding of the effect of diabetes? Would it not be required to develop a diabetic population and what considerations would be required to develop this population?
  3. From the title of the manuscript one would expect a more mechanistic review of the effect of diabetes on CYP-transporter interplay, which the reviewer found missing. For example, would PPARa/g and/or glucocorticoid receptor pathway be involved in this altered function? How does the interplay take place and if there is any biomarker to investigate such interaction, which gets affected by diabetes? 

Minor comments:

  1. The reference is missing for the sentence in line# 58-60. 
  2. Line# 102, we were talking about mdr1a and mdr1b, then here only Pgp protein was mentioned. Which MDR protein was found altered and if the authors have any hypothesis for this observation?
  3. Line# 189-190, "Vmax/Km values of phenytoin" for which enzyme/ trasnporter/ metabolism or transport process? Please be specific. 
  4. what is the reason for altered blood flow consideration in the PBPK model, as presented in Table 1?

Author Response

Please see the response as attached.

Round 2

Reviewer 1 Report

I am satisfied with the improvements made in the revised version of the paper.

Author Response

Thanks very much for your kind work and consideration on our paper.

Reviewer 3 Report

The authors addressed the major concerns and critique.

Author Response

(The authors gave the same response as above.)

Reviewer 4 Report

The authors need to carefully review the writing style in the manuscript, as it conveys incomplete, nonspecific and sometimes confusing information. 

For e.g. from the response: 

  1. " The review stated that alterations in expressions and functions of transporters and CYP450s by diabetes were dependent on specific isoforms and tissues" - Specific isoforms of what? Transporters or enzymes? I would assume the authors meant, specific isoforms of CYP enzymes. 
  2. " We raised hypothesis that PBPK model could be used to accomplish disposition of drugs by diseases including diabetes" - Do the authors meant to convey " We hypothesized that PBPK model could be used to predict disposition of drugs in diabetes" ? Please check the sentence configuration carefully. Also, I think the authors should be talking specifically about diabetes, for e.g in the above sentence "accomplish disposition of drugs by diseases including diabetes" - the authors have not tested/ showed evidence of PBPK model for other disease conditions. So please be specific. 

Minor point 2. The reviewer is aware of mdr1a/b genes and proteins in rodents. From line# 109 to 113, please read carefully. The authors first mention that in rats, mdr1a is increased but mdr1b is decreased, then again says both mdr1a/b is significantly decreased, then mentions that immunoblotting showed that Pgp expression is decreased. There is a lot of confusion in this entire sentence. To avoid such complication, the authors can provide the final outcome, " Pgp expression decreased". Additional sentences related to mRNA expression is creating more confusion. 

Minor point 3. The sentence is still incorrect in line# 235. The authors mention in the response " Vmax/Km values of phenytoin" should be parameters for metabolism" - but this is not in the corrected version. How will the readers know what Vmax they are dealing with!?

Minor point 4. Blood flow in multiple organs were considered different in the table. Please provide reference for each organ or tissue. Or mention clearly what was the hypothesis for altered blood flow in Duodenum, Jejunum, Ileum wall, portal vein, Hepatic artery, liver, renal flow, along with liver weight. Are all these parameters independent or correlated? Or were they identified uniquely?

Author Response

Point 1: " The review stated that alterations in expressions and functions of transporters and CYP450s by diabetes were dependent on specific isoforms and tissues" - Specific isoforms of what? Transporters or enzymes? I would assume the authors meant, specific isoforms of CYP enzymes. 

Response 1:

        Thank your suggestion. The specific isoforms are for CYP enzymes and transporters. We will rewrite this sentence as following.

“Alterations in expressions and functions of transporters and CYP450s by diabetes are dependent on specific isoforms (CYP450s and transporters) and tissues”

Point 2: " We raised hypothesis that PBPK model could be used to accomplish disposition of drugs by diseases including diabetes" - Do the authors meant to convey " We hypothesized that PBPK model could be used to predict disposition of drugs in diabetes" ? Please check the sentence configuration carefully. Also, I think the authors should be talking specifically about diabetes, for e.g in the above sentence "accomplish disposition of drugs by diseases including diabetes" - the authors have not tested/ showed evidence of PBPK model for other disease conditions. So please be specific. 

Response 2:

        Thank your suggestion we have rewritten these sentence as follows:

“We thought that PBPK model could be used to accomplish prediction of drug disposition under diabetic status”

Point 3 (Minor point 2.): The reviewer is aware of mdr1a/b genes and proteins in rodents. From line# 109 to 113, please read carefully. The authors first mention that in rats, mdr1a is increased but mdr1b is decreased, then again says both mdr1a/b is significantly decreased, then mentions that immunoblotting showed that Pgp expression is decreased. There is a lot of confusion in this entire sentence. To avoid such complication, the authors can provide the final outcome, " Pgp expression decreased". Additional sentences related to mRNA expression is creating more confusion. 

Response 3:

        Thank your suggestion. The report mentioned in the line# 109 to 113 researched two diabetic progressions in STZ-induced diabetes Sprague-Dawley rats, one is 5-week diabetes and another is 8-week diabetes. Mdr1a/b genes and P-gp protein have different manners in different diabetic progressions. Moreover, in different rat species, the alternations of Mdr1a/b genes and P-gp protein are also different in different researches.

In rat. P-gp is encoded by two genes(mdr1a and Mdr1b). Usually, mRNA of Mdr1a is higher than that of Mdr1b. That paper showed that alterations in Mdr1a and mdr1b by diabetes were dependent on species isoform and diabetic progression. Sometimes, expressions of P-gp was not in line with mRNA of its genes, indicating that regulation of P-gp protein expression occurred at post-transcriptional levels, which may explain that contrast results on expression of P-gp protein and its mRNA.        

Point 4 (Minor point 3.): The sentence is still incorrect in line# 235. The authors mention in the response " Vmax/Km values of phenytoin" should be parameters for metabolism" - but this is not in the corrected version. How will the readers know what Vmax they are dealing with!?

Response:

Thank your suggestion. Reference did not mention whether Vmax and Km was enzymic. The data were estimated using plasma concentration of phenytoin following dose using M-M equation dC/dt=VmaxC/(Km+C). Theoretically, Vmax and Km reflected enzymic parameters.

Point 5 (Minor point 4.): Blood flow in multiple organs were considered different in the table. Please provide reference for each organ or tissue. Or mention clearly what was the hypothesis for altered blood flow in Duodenum, Jejunum, Ileum wall, portal vein, Hepatic artery, liver, renal flow, along with liver weight. Are all these parameters independent or correlated? Or were they identified uniquely?

Response:

        Thank your suggestion. The parameters mentioned in question are all from the previously published articles in this laboratory [11], as cites in table 1. The initial reference will give as following [1][2][3].

[1] Hill M A, Larkins R G. Alterations in distribution of cardiac output in experimental diabetes in rats [J]. Am J Physiol, 1989, 257(2 Pt 2): H571-80.

[2] Lucas P D, Foy J M. Effects of experimental diabetes and genetic obesity on regional blood flow in the rat [J]. Diabetes, 1977, 26(8): 786-92.

[3] Granneman JG and Stricker EM. Food intake and gastric emptying in rats with streptozotocin-induced diabetes. Am J Physiol, 1984, 247:R1054-1061.

Once again, thank you very much for your comments and suggestions.

Round 3

Reviewer 4 Report

Thank you for the revision.